# Augmented PIN Authentication through Behavioral Biometrics

**DOI:** 10.3390/s22134857

**Published:** 2022-06-27

**Authors:** Matteo Nerini, Elia Favarelli, Marco Chiani

**Affiliations:** 1Department of Electrical and Electronic Engineering, Imperial College London, London SW7 2AZ, UK; m.nerini20@imperial.ac.uk; 2Department of Electrical, Electronic and Information Engineering (DEI), University of Bologna, 40136 Bologna, Italy; elia.favarelli@unibo.it

**Keywords:** behavioral biometrics, motion sensors, Machine Learning, cyber security, Personal Identification Number

## Abstract

Personal Identification Numbers (PINs) are widely used today for user authentication on mobile devices. However, this authentication method can be subject to several attacks such as phishing, smudge, and side-channel. In this paper, we increase the security of PIN-based authentication by considering behavioral biometrics, specifically the smartphone movements typical of each user. To this end, we propose a method based on anomaly detection that is capable of recognizing whether the PIN is inserted by the smartphone owner or by an attacker. This decision is taken according to the smartphone movements, which are recorded during the PIN insertion through the built-in motion sensors. For each digit in the PIN, an anomaly score is computed using Machine Learning (ML) techniques. Subsequently, these scores are combined to obtain the final decision metric. Numerical results show that our authentication method can achieve an Equal Error Rate (EER) as low as 5% in the case of 4-digit PINs, and 4% in the case of 6-digit PINs. Considering a reduced training set, composed of solely 50 samples, the EER only slightly worsens, reaching 6%. The practicality of our approach is further confirmed by the low processing time required, on the order of fractions of milliseconds.

## 1. Introduction

The PIN is a numeric password commonly used to provide user authentication together with novel alternatives such as pattern lock and fingerprints. Typical uses of the PIN include screen unlocking on mobile devices, user authentication on computers, and secure access to specific services, such as banking systems. However, this authentication method can be subject to three types of cyber attacks. Firstly, an attacker could learn the PIN through traditional attacks such as phishing and shoulder surfing [1]. Secondly, smudge attacks are possible when the PIN is inserted on the touchscreen of a mobile device [2]. This kind of attack relies on the smudge left behind by the user’s fingers on the touchscreen to infer information on the typed digits. Finally, a variety of side-channel attacks have been investigated for mobile devices [3,4,5,6,7]. Here, the side-channel information is given by the movements done by the user while inserting the PIN. These movements can be acquired by the built-in motion sensors and used to infer the secret digit combination. For example, in [7] it has been shown that a 4-digit PIN can be correctly recovered with an 84% success within 20 tries.

Several solutions have been proposed to make a PIN, or an alphanumeric password, more robust against these kinds of attacks. In some works, the PIN is entered using touch gestures substituting the keystrokes, making it more secure against shoulder surfer and side-channel attacks [1,8]. In [3], the effectiveness of side-channel attacks is reduced by adding Gaussian noise to the motion sensor data. However, this data perturbation also affects the accuracy of the sensors and their utility. In [9], the actual PIN digits are inserted alternated with misleading values, on the basis of the screen brightness. Since the screen brightness value is not visible to screen recording techniques, this authentication method is resilient against side-channel and spyware-based attacks. In [10], the user is requested to draw the digits of the PIN on the touchscreen. In this way, the user’s drawing traits are used as a further authentication measure, beyond the secrecy of the PIN. Finally, keystroke dynamics information, describing the person’s typing rhythm, can be used to enhance the security of alphanumeric passwords [11,12], or to provide free-text authentication [13].

Other studies proposed authentication methods completely based on behavioral biometrics, which are recurrent patterns typical of each user behavior [14,15,16]. Most solutions utilize wrist-worn devices, such as smartwatches. In [17,18,19], these devices are used to collect wrist movements while the user performs a specific gesture. Thus, the identity of the person wearing the device is verified from these motion data. In [20,21] wrist movements are analyzed jointly with mouse and keystroke activities. The correlation between these activities is used to provide Continuous Authentication (CA). CA can be provided also on the basis of the behavioral traits contained in the movements recorded by smartphones [22,23]. Here, the user is authenticated while performing daily life activities such as sitting, walking, and running. In [24], CA on mobile devices is achieved by exploiting app-usage information. However, authentication methods solely based on behavioral biometrics might be not enough accurate due to the irregular nature of the movements [25].

In practical authentication, we can identify two main requirements that need to be met. Firstly, a very high level of security is required, i.e., only the smartphone owner should be able to access the service. Because of this requirement, authentication methods purely based on behavioral biometrics are not reliable enough to be suitable in practical applications. Secondly, the smartphone owner should take little effort to access the service. This is a crucial requirement since many authentication systems are accessed multiple times every day, such as screen unlocking. For this reason, related work aiming at strengthening the PIN security with additional user actions, e.g., gestures or drawings, may fail to satisfy this requirement [9,10]. In this work, we aim at providing a highly secure authentication method that satisfies both these two challenges. This is realized by including the traditional PIN in the authentication and capturing sensor data while the user is inserting it. Thanks to this methodology, for an attacker it becomes harder to access the system since it needs to both steal the PIN and be allowed by the anomaly detector. At the same time, the process does not require additional effort from the user except entering the usual PIN.

In this work, we propose a novel authentication method that uses behavioral biometrics to increase the PIN security. Assuming that a correct PIN is inserted in a smartphone, our method verifies whether the user who typed the PIN was the actual smartphone owner or an attacker. To this end, the smartphone movements are recorded during the PIN insertion through built-in motion sensors, present in every smartphone. Then, an anomaly detection-based system evaluates whether these movements represent an inlier (i.e., the smartphone owner typed the PIN), or an anomaly (i.e., an attacker typed the PIN). We implement and test our authentication method using four common anomaly detection algorithms: Principal Component Analysis (PCA), Kernel Principal Component Analysis (K-PCA), One-Class Support Vector Machine (OC-SVM), and Local Outlier Factor (LOF). Finally, their performances are assessed and compared on the basis of real data. More precisely, we exploit the fact that a user needs to type *N* times in the case of an *N*-digit PIN. This offers *N* samples of the motion sensors that can be combined to increase the performance of the anomaly detectors as *N* increases. At the same time, this process does not require further actions from the users, differently from related works on behavioral biometrics. Furthermore, since the movements are recorded by the smartphone, we do not assume the presence of a smart bracelet to acquire behavioral biometrics (to promote reproducible research, the simulation code is available at: https://github.com/matteonerini/augmented-pin-authentication, accessed on 27 May 2022).

The rest of the paper is organized as follows. In Section 2, we present our anomaly detection-based authentication method and the sensor features that are involved. In Section 3, we briefly describe the four anomaly detection algorithms considered in this paper. In Section 4, we describe the real-world data acquisition process and assess the obtained accuracy through real experiments. In Section 5, we highlight the reasons behind the obtained results and possible applications of our research. Finally, Section 6 contains the concluding remarks.

Throughout the paper, vectors and matrices are denoted with bold lower and bold upper letters, respectively. Scalars are represented with letters not in bold font. ${\left(\cdot \right)}^{T}$ stands for transposition and ${∥\cdot ∥}_{2}$ is the $\ell_{2}$-norm of a vector.

## 2. Methodology

Let us assume that an *N*-digit PIN is used as an authentication method on a smartphone device. This PIN can guarantee authentication to access any service of practical interest, i.e., screen unlocking or mobile banking. The idea behind this authentication method is that only the smartphone owner knows the PIN. Thus, if an attacker could steal such a combination, he would be able to break the authentication. In this paper, we strengthen the security of this authentication method by verifying whether the correct PIN has been inserted by the actual smartphone owner. This is realized assuming that users hold their smartphone and type the PIN in a personal manner. As a result, different users can be distinguished on the basis of the smartphone movements they produce while inserting the PIN.

### 2.1. Feature Selection

To improve the authentication security, we assume that the smartphone movements are captured for each digit composing the PIN, i.e., for each keystroke. Movements are sampled from built-in sensors and represented by a total of *D* features. Three categories of sensors are supported by Android OS: motion sensors, environmental sensors, and position sensors [26]. For our purposes, we consider only motion sensors, whose values are provided according to a coordinate system defined with respect to the device screen [26]. More precisely, the $\hat{i}$ axis is horizontal and points to the right, the $\hat{j}$ axis is vertical and points upward and the $\hat{k}$ axis is perpendicular to the screen of the device, as shown in Figure 1. We select six relevant sensors to fully characterize the smartphone movements, which are described below:The Accelerometer measures the total acceleration, given in $m/s^{2}$, experienced on the three axes $\hat{i},\hat{j},\hat{k}$.The Gravity sensor gives the components of the gravitational acceleration *g* in $m/s^{2}$ along each direction $\hat{i},\hat{j},\hat{k}$.The Gyroscope measures the angular speed, in rad/s, around the three device axes $\hat{i},\hat{j},\hat{k}$. The rotation is positive in the counter-clockwise direction.The Linear Acceleration indicates the acceleration experienced along each device axis $\hat{i},\hat{j},\hat{k}$, without the gravity contribution.The Rotation Vector is composed of three dimensionless components and it represents the rotations of the device $\hat{i},\hat{j},\hat{k}$ axes with respect to the east, the geomagnetic north, and the zenith, respectively.The Orientation sensor returns an array of three angles: the *azimut*, i.e., is the angle between the geomagnetic north direction and the device $\hat{j}$ axis; the *pitch*, i.e., is the angle of rotation around the $\hat{i}$ axis; and the *roll*, which is the angle of rotation around the $\hat{j}$ axis.

Each of these six sensors returns three values, providing a total of 18 values. Among them, the ones referring to the cardinal directions are discarded because they are not relevant to our scope. In particular, we do not consider the components of the Rotation Vector taken with reference to the north and east directions, and the *azimut* value of the Orientation sensor. In addition, we append to the features set the value of the pressed digit, and a further value *M*, defined as $M=\sqrt{pitch^{2}+roll^{2}}$. In conclusion, a total of D=17 features have been used to represent the smartphone movement associated with each keystroke.

### 2.2. Anomaly Detection-Based Authentication

Our ultimate goal is to verify whether a correct *N*-digit PIN has been inserted by the actual smartphone owner or by an attacker. To this end, we propose an anomaly detection-based method composed of three stages, as represented in Figure 1.

Firstly, in the *sampling* stage, the *N*-digit PIN is inserted through *N* keystrokes by the user, who can be the smartphone owner or an attacker. At each keystroke, the considered motion sensors are sampled and the *D* features are collected. We denote with $z^{\left(n\right)}$ the vector containing the *D* features corresponding to the *n*-th PIN digit, with n=1,⋯,N.

After that, in the *anomaly detection* stage, *N* anomaly detectors are independently computed. The *n*-th anomaly detector $f_{AD}\left(\cdot \right)$ evaluates whether the *n*-th digit has been inserted by the actual smartphone owner on the basis of the feature vector $z^{\left(n\right)}$. The function $f_{AD}\left(\cdot \right)$ represents, in general, any anomaly detector. In this study, we consider four different anomaly detection algorithms and we compare their performance: PCA, K-PCA, OC-SVM, and LOF. We denote with the scalar $s^{\left(n\right)}$ the output of the *n*-th $f_{AD}\left(\cdot \right)$, representing the anomaly score of $z^{\left(n\right)}$. A low $s^{\left(n\right)}$ suggests that the *n*-th digit has been inserted by an attacker, i.e., it is an anomaly. On the contrary, a high $s^{\left(n\right)}$ suggests that the *n*-th digit has been inserted by the actual smartphone owner, i.e., it is an inlier.

Finally, the *N* anomaly scores are combined in the *combining* stage to obtain a more reliable anomaly score *s*. The combining operation consists in averaging the *N* scores; thus, *s* is obtained as
$$s=\frac{1}{N}\sum_{n=1}^{N}s^{\left(n\right)}.$$

Based on the value of *s*, the final decision $\hat{y}$ is taken, with $\hat{y}\in \left\{\mathrm{inlier},\mathrm{anomaly}\right\}$. If *s* is below a certain threshold $s_{\mathrm{th}}$, the whole PIN is considered inserted by an attacker, i.e., $\hat{y}=\mathrm{anomaly}$. In this case, the authentication system could ask the user to reinsert the PIN in practical developments. Otherwise, the smartphone owner’s identity is verified, i.e., $\hat{y}=\mathrm{inlier}$, and the user is allowed to access the requested service. Intuitively speaking, in this third stage, we exploit the diversity offered by the digits composing the PIN to enhance the performance of our authentication method. Thus, longer PINs offer greater authentication accuracy.

## 3. The Adopted Anomaly Detection Algorithms

In this study, we explore the performance of four ML techniques commonly used for unsupervised anomaly detection tasks [27,28,29], implemented in Python with the Scikit-learn library [30]. In this section, we briefly recall the algorithms and the chosen hyperparameters, which have been tuned through 5-fold cross validation [31].

In the following, $\bar{X}\in R^{N_{x}\times D}$ denotes the training set matrix, whose rows are the $N_{x}$ inliers training points, $\bar{Y}\in R^{N_{y}\times D}$ contains the $N_{y}$ inliers test points, and $\bar{U}\in R^{N_{u}\times D}$ contains the $N_{u}$ anomalies test points. Let us define the offset μ as the row vector containing the column-wise mean of the matrix $\bar{X}$, and the scaling factor σ as the row vector containing the column-wise standard deviation of the matrix $\bar{X}$. Before proceeding with the anomaly detection, the features in the matrices $\bar{X}$, $\bar{Y}$, and $\bar{U}$ are centered and normalized by subtracting to each row the offset μ and dividing each row element-wise by the scaling factor σ. The resulting data matrices are denoted by X, Y, and U.

### 3.1. Principal Component Analysis

PCA is a technique that realizes linear dimensionality reduction by mapping the training data from the *D*-dimensional feature space $R^{D}$ into a subspace $R^{P}$, where D=17 in our problem, and P<D is the number of principal components selected. The subspace is determined such that it minimizes the error (defined as Euclidean distance) between the data in the feature space and their projection in the selected subspace [32]. In more detail, to find the best subspace to project the training data, we need to evaluate the D×D sample covariance matrix
$$\Sigma_{x}=\frac{X^{T}X}{N_{x}-1}.$$

By eigenvalue decomposition, $\Sigma_{x}$ can be factorized as $\Sigma_{x}=V_{x}\Lambda_{x}{V_{x}}^{T}$, where $V_{x}$ is an orthonormal matrix whose columns are the eigenvectors, while $\Lambda_{x}$ is a diagonal matrix containing the *D* eigenvalues. The eigenvalues’ magnitude represents the importance of the direction pointed by the relative eigenvector. Let us assume that the eigenvalues in $\Lambda_{x}$ are ordered in descending order and that the eigenvectors in $V_{x}$ are ordered accordingly. To select the first *P* components of the subspace, the matrix $V_{P}\in R^{D\times P}$ containing the first *P* columns of $V_{x}$ needs to be considered. Hence, the projection into the subspace is obtained by multiplying the data by $V_{p}$, i.e., $X_{P}=XV_{P}$, $Y_{P}=YV_{P}$, and $U_{P}=UV_{P}$. To evaluate the error between the projected points and the original ones it is necessary to reconstruct the data in the original feature space, i.e., $\tilde{X}=X_{P}V_{P}^{T}$, $\tilde{Y}=Y_{P}V_{P}^{T}$, and $\tilde{U}=U_{P}V_{P}^{T}$.

After the reconstruction, it is possible to evaluate the anomaly score as the opposite of the Euclidean distance between the original data and reconstructed data. Thus, given a generic point $z\in R^{D}$, its anomaly score is given by
(1)$$s=-{∥z-\tilde{z}∥}_{2},$$
where $\tilde{z}=zV_{P}V_{P}^{T}$ is the reconstructed version of z. Since the PCA has been trained on inlier samples, a z with a high *s* (i.e., close to zero) is likely to be an inlier. Conversely, a low *s* indicates the presence of an anomaly. The value of *P* is a hyperparameter that needs to be optimized. A too high *P* would yield a good reconstruction quality for all the samples (both inliers and anomalies), while a too low *P* would significantly compress the feature space lowering the reconstruction quality. In both these extreme cases, the task of differentiating inliers and anomalies would become hard. In our PCA implementation, through 5-fold cross validation, we verified that the subspace dimensionality P=10 ensures the optimal trade-off between reconstruction quality and compression.

### 3.2. Kernel Principal Component Analysis

Differently from PCA, K-PCA firstly maps the data with a non-linear function, named kernel, then applies the standard PCA to perform a linear dimensionality reduction in the new feature space. As a result, such dimensionality reduction becomes non-linear in the original feature space. A crucial point in K-PCA is the selection of the kernel that leads to linearly separable data in the new feature space. In [33], when the data distribution is unknown, the Radial Basis Function (RBF) kernel is proposed as a good candidate to accomplish this task. With this kernel, a generic point $z\in R^{D}$ is mapped in the vector $k^{\left(z\right)}=\left[K_{1}^{\left(z\right)},K_{2}^{\left(z\right)},\cdots ,K_{N_{x}}^{\left(z\right)}\right]$. Specifically, given the vector z, we can apply the RBF as
$$K_{i}^{\left(z\right)}=\exp\left(-\gamma {∥z-x_{i}∥}_{2}^{2}\right),$$
with $i=1,2,\cdots ,N_{x}$. In this transformation, γ is the kernel coefficient controlling the width of the Gaussian function, $x_{i}$ is the *i*-th row of X, and $K_{i}^{\left(z\right)}$ is the *i*-th component of the point z in the kernel space. Remapping all the data in the kernel space, we obtain the following matrices: $K_{x}\in R^{N_{x}\times N_{x}}$ for training, $K_{y}\in R^{N_{y}\times N_{x}}$ for testing on inliers, and $K_{u}\in R^{N_{u}\times N_{x}}$ for testing on anomalies.

Applying now the PCA to the new data sets, it is possible to perform non-linear dimensionality reduction from the original feature space to a subspace $R^{Q}$. Finally, the anomaly score can be again calculated as in (Equation 1). In our K-PCA implementation, the hyperparameters set through 5-fold cross validation are γ=1/17 and Q=8. The kernel coefficient γ influences the sensitivity to differences in feature vectors. A too large γ would cause overfitting, meaning that only points extremely close to the training set would be classified as inliers. Conversely, a too small γ would make it impossible to distinguish between inliers and anomalies. The meaning of *Q* is similar to the meaning of the parameter *P*, previously introduced for PCA.

### 3.3. One-Class Support Vector Machine

The OC-SVM algorithm has the objective of learning a close frontier delimiting a given training set X, as introduced by Schölkopf et al. in [34]. In this way, a new point is classified as an inlier if lying within this frontier, or as an anomaly otherwise. The main idea is to map the training data into a different feature space with a fixed transformation and to separate them from the origin with a maximum margin. We denote with ϕ(·) this fixed feature space transformation. Thus, the goal is to learn the weights w and the maximum margin ρ by minimizing the objective function as follows
$$\begin{array}{cc} \min_{w,\zeta_{i},\rho }\;\;\; & \frac{1}{\nu N_{x}}\sum_{i=1}^{N_{x}}\zeta_{i}-\rho +\frac{1}{2}{∥w∥}_{2}^{2}\\ s.t.\;\;\;\; & \left(w^{T}\phi \left(x_{i}\right)\right)\geq \rho -\zeta_{i},\;\forall i,\\ \; & \zeta_{i}\geq 0,\;\forall i, \end{array}$$
where $\zeta_{i}$ is the margin violation for the training point $x_{i}$. In the objective function to be minimized, the hyperparameter $\nu \in \left(0,1\right]$ controls the strength of the regularization term $\frac{1}{2}{∥w∥}_{2}^{2}$. Furthermore, it can be proven that ν is an upper bound on the fraction of training points outside the estimated frontier, and a lower bound on the fraction of support vectors [34,35]. Assuming that w and ρ solve the problem, the anomaly score is defined for a generic point z as
(2)$$s=\left(w^{T}\phi \left(z\right)\right)-\rho ,$$
which can be referred to as the signed distance between z and the separating hyperplane. Since the variables $\zeta_{i}$ penalize the objective function, *s* is likely to be positive for most training samples.

The aforementioned optimization problem can be solved through its dual [34], defined as
$$\begin{array}{cc} \min_{\alpha_{i}}\;\;\; & \frac{1}{2}\sum_{i=1}^{N_{x}}\sum_{j=1}^{N_{x}}\alpha_{i}\alpha_{j}K\left(x_{i},x_{j}\right)\\ s.t.\;\;\;\; & 0\leq \alpha_{i}\leq \frac{1}{\nu N_{o}},\;\forall i,\\ \; & \sum_{i=1}^{N_{o}}\alpha_{i}=1, \end{array}$$
where $a_{n}$ are defined such that $w=\sum_{i=1}^{N_{x}}\alpha_{i}\phi \left(x_{i}\right)$ and $K\left(x_{i},x_{j}\right)=\phi {\left(x_{i}\right)}^{T}\phi \left(x_{j}\right)$ is the kernel [36]. In this way, the anomaly score (Equation 2) of a sample z can be rewritten as
(3)$$s=\sum_{i=1}^{N_{x}}\alpha_{i}K\left(x_{i},z\right)-\rho .$$

In our OC-SVM implementation, we use an RBF kernel defined by
$$K\left(x_{i},x_{j}\right)=\exp\left(-\gamma {∥x_{i}-x_{j}∥}_{2}^{2}\right),$$
where $x_{i}$ and $x_{j}$ are two generic points [37]. Furthermore, the hyperparameters ν=0.1 and γ=1/17 have been chosen on the basis of 5-fold cross validation. As anticipated, ν is the hyperparameter controlling the strength of the regularization term $\frac{1}{2}{∥w∥}_{2}^{2}$ in the objective function. With a very low ν, the contribution of the regularization term would be negligible, and the weights w would be learned with no restriction. In this case, overfitting would be likely to occur. Conversely, with a too high ν, it would be harder to learn a meaningful frontier since ${∥w∥}_{2}^{2}$ would tend to zero.

### 3.4. Local Outlier Factor

The LOF algorithm identifies anomalies based on the local density of points within the dataset [38]. This unsupervised learning technique receives in input a set of points composed by the training set X, containing examples of inliers, and a new point z, which has to be classified as an inlier or anomaly. The main intuition is that the density of the samples around an anomaly, also called an outlier, should be significantly lower than the density around its neighbors. To formalize this concept, the *k*-distance of z, denoted as k-dist(z) has been introduced as the Euclidean distance between the point z and its *k*-th nearest neighbor [38]. The *k*-distance is used to define the reachability distance of z from $x_{i}$ as
$$reach-dist_{k}\left(z,x_{i}\right)=\max\left\{k-dist\left(z\right),d\left(z,x_{i}\right)\right\},$$
where $d(z,x_{i})$ denotes the Euclidean distance between z and $x_{i}$. Thus, the formal notion of density around z used in the LOF algorithm is given by the local reachability density of z, defined as
$$lrd_{k}\left(z\right)=\frac{k}{\sum_{x_{i}\in N_{k}\left(z\right)}reach-dist_{k}\left(z,x_{i}\right)},$$
where $N_{k}\left(z\right)$ denotes the set of the *k*-nearest neighbors of z. In other words, the local reachability density can be interpreted as the inverse of the average reachability distance of z from its *k* neighbors. Finally, the local reachability density of z is compared with those of the neighbors in the local outlier factor metric, defined as
$$lof_{k}\left(z\right)=\frac{\sum_{x_{i}\in N_{k}\left(z\right)}\frac{lrd_{k}\left(x_{i}\right)}{lrd_{k}\left(z\right)}}{k}.$$

According to this definition, $lof_{k}\left(z\right)$ is the average local reachability density of the *k* neighbors divided by the local reachability density of z.

A lof value close to 1, or less than 1, is an indicator that the observation is an inlier. On the other hand, a lof much greater than 1 indicates the presence of an anomaly, since the density of points around z is much less than the densities around its *k*-nearest neighbors. In our LOF implementation, the threshold which discriminates between inliers and anomalies is set to $\bar{lof}=1.5$, as in the original paper [38], and we select k=30 with 5-fold cross validation. Note that the higher the threshold $\bar{lof}$, the more samples are accepted as inliers. Moreover, the effect of the neighborhood size *k* is to determine the amount of local information to capture. Finally, the anomaly score of a sample z in the output of the LOF anomaly detector is
(4)$$s=-(lof_{k}\left(z\right)-1.5).$$

In this way, negative scores represent anomalies, while positive scores represent inliers.

The four considered anomaly detectors are summarized in Table 1. To summarize, the four considered anomaly detectors are characterized by different anomaly scores, as defined in (Equation 1), (Equation 3) and (Equation 4). These scores have all been designed such that high values correspond to inliers, while low values correspond to anomalies.

## 4. Experimental Methods and Results

In this section, we evaluate the performance of our authentication method. Firstly, we describe the data acquisition process that allowed us to build a dataset to train and test the proposed authentication method. Secondly, we measure the authentication accuracy and discuss the obtained numerical results.

### 4.1. Data Collection

To acquire the data, we developed a smartphone application able to sample the motion sensors when a user enters a digit in a numeric keypad. The application, developed with the Android Studio environment, has been designed to collect both the training set, that instructs the anomaly detection algorithms, and the test set, to evaluate their detection performance. The user interface consists of a single screen exhibiting a numeric keypad, as represented in Figure 1. The recorded data is organized by the application into a table stored in a Comma-Separated Values (CSV) file. When a digit in this keypad is pressed, the considered motion sensors are recorded together with the value of the pressed digit, and a row is added to the CSV file. In our study, the sensors are sampled only in correspondence with a keystroke. This is realized in Android Studio by triggering the sampling operation through the *android:onClick* attribute of the *<Button>* element in the XML layout of the application. Thanks to this sampling strategy, particular data preprocessing is not necessary. As anticipated in Section 2, each row of the obtained CSV file, representing a sample, contains 17 numerical scalar values: one representing the pressed digit, and 16 sampled by the considered motion sensors. Once the typing session is terminated, the application returns the CSV file labeled with the user identifier of the smartphone owner.

We recruited 12 volunteer students, standard smartphone users, who installed the dedicated application on their smartphones. All the devices were Android, running an updated version of the operative system (7.0 Nougat or later). Each student was asked to type a list of 500 digits randomly generated in the numeric keypad shown on the screen, while naturally holding their smartphone with one hand. The random digits generator took care that the digits entered by the users were equally distributed (50 samples of each digit per student). At the end of the typing session, not all the students typed exactly 500 digits, but all of them typed at least 470 digits. Thus, the dataset was randomly cleaned in order to have exactly 470 samples per student. Finally, the obtained dataset has been preprocessed by normalizing to zero mean and unit variance every feature. This last step has the objective to make each feature equally important, and it is particularly useful when dealing with ML algorithms [31].

The resulting dataset was used to train and test 12 authenticators in an unsupervised manner, one for each student. The 12 training sets were composed of 90% of the data, i.e., 423 samples of the same student for each set. These sets were used to choose the hyperparameters of the ML models via 5-fold cross validation, and for the final trainings. The final trainings were carried out considering four different training set sizes, to investigate their impact on the authentication accuracy: $N_{x}=50$, $N_{x}=100$, $N_{x}=200$, and $N_{x}=400$. The 12 inlier test sets included the remaining 10% of the data, i.e., $N_{y}=47$ samples of the same student for each set. Finally, the anomaly test set for each student was given by the inlier test sets of all the other students. Thus, the 12 anomaly test sets were composed of $N_{u}=47\times \left(12-1\right)=517$ samples.

### 4.2. Testing Strategy

To evaluate the performance of our authentication method, we consider 12 one-vs-all authentication problems. In each of these problems, $N_{x}$ training samples inserted by a specific student are used to train an anomaly detector $f_{AD}\left(\cdot \right)$ in an unsupervised manner. Then, to test the validity of an *N*-digit PIN, we feed *N* trained anomaly detectors with *N* samples $\left\{z^{\left(1\right)},z^{\left(2\right)},\cdots ,z^{\left(N\right)}\right\}$ (see also Figure 1). The *N* samples are all taken either from the corresponding inlier test set (assuming the PIN has been inserted by the smartphone owner), or the anomaly test set (assuming the PIN has been inserted by an attacker). Exploiting the Monte Carlo method, we randomly generate 500 different combinations of *N* samples, without replacement, from both the inlier test set and the anomaly test set. The resulting $\hat{y}$ are checked to verify whether the authenticator decisions are correct. Finally, the performances are averaged over the 12 one-vs-all authentication problems.

### 4.3. Numerical Results

We first evaluate the accuracy of our authentication method in terms of Receiver Operating Characteristic (ROC) and Area Under Curve (AUC). A ROC is a graphical representation of all the possible working points of a binary classifier. This curve is derived by plotting the true positive rate TPR versus the false positive rate FPR obtained by varying the discrimination threshold $s_{\mathrm{th}}$. Figure 2 illustrates the ROCs obtained by the four considered anomaly detectors, trained with $N_{x}=400$, and for three different PIN lengths: N=3, N=4, and N=6. In addition, for each curve, we report the AUC metric, defined as the area under it. First of all, we notice that longer PINs allow more accurate authentications. Indeed, the diversity order of our authentication method is equal to the PIN length *N*. Second, among the considered anomaly detection algorithms, PCA is the most performing in terms of AUC. It also outperforms the other algorithms approximately in every working point of the ROC of practical interest.

Now, we investigate how reduced training set sizes $N_{x}$ affect the performance of our authenticator. To this end, we consider two performance metrics. Firstly, the EER is given by the common value of the true positive rate and the true negative rate when they are equal. This is graphically represented for each ROC in Figure 2 by the intersection point between the ROC and the dashed bisector. Secondly, we consider the Maximum Balanced Accuracy (MBA), given by the maximum value assumed by the balanced accuracy. In turn, the balanced accuracy for a specific working point is defined as the average between the true positive rate and the true negative rate, i.e.,
$$BA=\frac{TPR+TNR}{2}.$$

Note that the balanced accuracy corresponds to the accuracy when the number of positives is equal to the number of negatives in the test set. The working point corresponding to the MBA of each ROC is graphically identified with a circle in Figure 2. The metrics EER and MBA are reported in Figure 3 for the four training set sizes considered. As expected, both 1−EER and the MBA increase with the PIN length *N*, and PCA is the best algorithm in all cases. In addition, we observe that the performance degrades only slightly when reduced training sets are used. For every PIN length *N* considered, $N_{x}=50$ samples are sufficient to train a well-performing authenticator, especially in the case of PCA. This characteristic further confirms the practical feasibility of our authentication method. It is worth noticing that typical training set dimensions adopted in the related works are greater than a hundred samples [10,11,12,16].

Since the behavior of users is not necessarily repetitive, false negatives might be detected by the authentication system. This means that the smartphone owner might experience a denial of access. To analyze the possible occurrence of denial of access to the user, we consider that the authentication system allows the user to reinsert the PIN multiple times in the case of access rejection. In Figure 4, we report the denial of access probabilities as a function of the number of consecutive attempts of PIN insertion. Here, we assume that the attempts are all correct and independent and that the anomaly authenticators are working in the EER working point on the ROC curve. In the case of 3-digit PINs, the denial of access probability after four correct attempts is approximately $10^{-5}$ when PCA is considered. If the user inserts the PIN on average 10 times every day, this event will occur on average once every 27 years. Since this interval of time is significantly beyond the expected lifetime of smartphone devices, this analysis confirms the practicality of our authentication method. To avoid denial of access, in practical scenarios a second authentication technique could be activated by the system after multiple rejections, e.g., a pre-established personal secret question could be asked.

Lastly, we analyze if the proposed authentication method is suitable to work in real-time by measuring its processing time. Since the authentication process needs to work in real-time, the testing phase of the algorithms employed should require a very low processing time. The experiments have been run on a Central Processing Unit (CPU) Intel Xeon, with a clock frequency of 2.20 GHz. In Table 2, we report the resulting processing times for the different algorithms, as a function of the PIN length *N*. We notice that the processing time mostly depends on the utilized algorithm, while we observe only a slight dependence on the PIN length. The testing time for all the four algorithms is below 1 ms, making them suitable for working in real-time. Furthermore, PCA is the detection algorithm with both the best accuracy and the lowest processing time, approximately 0.1 ms for all the considered PIN lengths. Note that this processing time is highly inferior to the interval of time needed to recognize a physical activity in purely behavioral biometric-based authentications, which is typically a few seconds [9,22].

## 5. Discussion

The fundamental assumption behind this paper, and in general behind all the studies on behavioral biometrics, is that each user has a personal way to interact with their devices. In our case, we assume that the users hold their smartphone and type the PIN in a personal manner. To validate this assumption, we carried out extensive experiments with real-world data collected on volunteers. The numerical results corroborate the validity of the assumption that each user has a personal way to interact with their devices. To observe the personal traits of the users, we asked our recruited students to naturally hold their smartphones with one hand. This was needed in order to capture significant movement variations with the build-in sensors. For instance, if the smartphone is lying on a table while the PIN is inserted, no behavioral biometric could be observed from the motion sensors and the PIN security could not be improved. In the case of practical development, this limitation could be overcome by asking the user to always insert the PIN in their natural way taking care that the smartphone is not in contact with external objects.

To explain this high performance obtained by our authentication method employing PCA, we inspect the dataset after applying dimensionality reduction through PCA. In Figure 5, we report the entire dataset, composed of 470 samples per student, projected along its first three principal components. Here, the samples corresponding to each user, identified by a unique color, are distributed in well-defined clusters, even when observed with a very limited dimensionality, i.e., P=3. Because of this property, PCA can distinguish the samples entered by the different students, obtaining a high authentication accuracy. In fact, we calculated that the total percentage of variance explained by the first P=10 principal components is 97.8% The reason behind the high performance of the PCA detector is that sample points of the same user are clustered in clouds. However, failure cases for the method are given by those points that are far from the center of their cluster. These points are easily misdetected by the method and may represent false positives or false negatives. The underlying cause of these failure cases is that the behavior of a user is not necessarily repetitive. In fact, a user may assume an unusual pose for only a few samples, which are represented by points far from the center of the user cluster.

With the widespread use of smartphone applications, there is an increasing need to protect the user’s privacy and security. The main applications that could benefit from our authentication system can be classified into four categories. First of all, screen unlocking is the most common security method adopted in smartphones. It is realized through the insertion of a PIN on a numeric keypad or a pattern on a grid of points. Thus, our authentication method can greatly contribute to strengthening the security of the PIN when used for screen unlocking purposes. Second, social network applications are a popular target of attackers willing to impersonate the device owner. In mobile devices, this risk is particularly high since users tend to be constantly logged in to the application, to avoid inserting the login credential at every access. A fast authentication method for these applications could be implemented through a short PIN strengthen by our anomaly detection-based method. Third, multimedia data such as photos and videos can be protected by creating secure folders on modern smartphones. To access such secure folders a PIN is commonly employed. However, given the paramount importance of preserving users’ privacy, the authentication method proposed in this paper could increase the PIN security to access these secure folders in smartphones. Lastly, transaction applications such as credit card and bank applications are ubiquitous nowadays. In recent years, virtual banks and home banking services increased their popularity, giving the user the possibility to perform transactions by solely using the smartphone. Before each transaction, our method could be employed to provide stronger user authentication, on top of the widely utilized numeric PINs.

## 6. Conclusions

In this study, we strengthen the security of *N*-digit PIN authentication on mobile devices by using behavioral biometrics. To this end, we propose a novel method to verify whether the correct PIN has been inserted by the actual smartphone owner or by an attacker. This method is based on 17 features extracted from the built-in smartphone motion sensors, and on the assumption that each user enters the PIN in a personal manner, learnable with anomaly detection algorithms. We implement our authentication method by comparing the performance of four different anomaly detectors: PCA, K-PCA, OC-SVM, and LOF.

Numerical results show that with PCA it is possible to achieve an EER as low as 5% in the case of 4-digit PINs, and 4% in the case of 6-digit PINs. Thus, in comparison to solely using the PIN as a security measure, the contribution brought by our authentication method can be summarized as follows. If only the PIN is used as an authentication method, an attacker who successfully stole the PIN is rejected by the system in the 0% of the cases. Conversely, when our method is employed, an attacker who successfully stole the PIN is rejected in the 96% of the cases considering a 6-digit PIN. Furthermore, the performance only slightly decreases when the training set size is reduced from 400 to 50 samples. The practicality of our approach is confirmed by the low processing time required, in the order of fractions of milliseconds for PCA. Compared to a purely PIN-based authentication, the improvement brought by our approach can be summarized as follows. An attacker, that would successfully authenticate by knowing the PIN, is not authenticated in the 96% of cases with our approach.

## Figures and Tables

**Figure 1 sensors-22-04857-f001:**
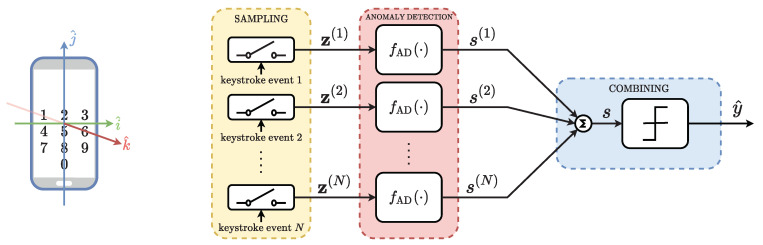
Anomaly detection-based authentication block diagram.

**Figure 2 sensors-22-04857-f002:**
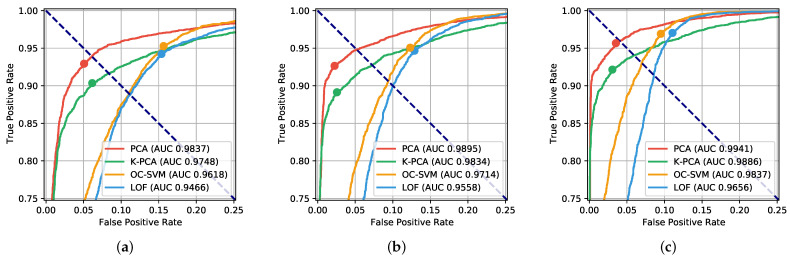
ROCs and AUC for different PIN lengths *N*. Working points corresponding to the MBA are marked with circles. (**a**) 3-digit PIN; (**b**) 4-digit PIN; (**c**) 6-digit PIN.

**Figure 3 sensors-22-04857-f003:**
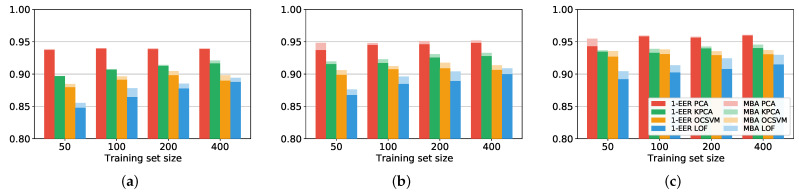
EER and MBA for different training set sizes $N_{x}$ and PIN lengths *N*. (**a**) 3-digit PIN; (**b**) 4-digit PIN; (**c**) 6-digit PIN.

**Figure 4 sensors-22-04857-f004:**
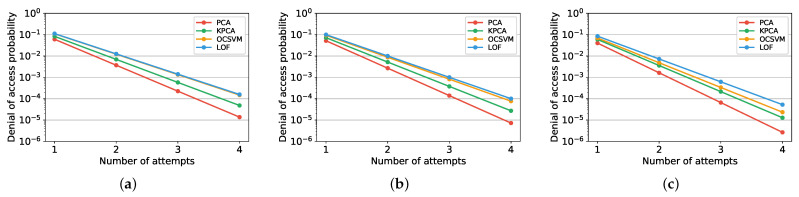
Denial of access probability for different number of consecutive attempts and PIN lengths *N*. (**a**) 3-digit PIN; (**b**) 4-digit PIN; (**c**) 6-digit PIN.

**Figure 5 sensors-22-04857-f005:**
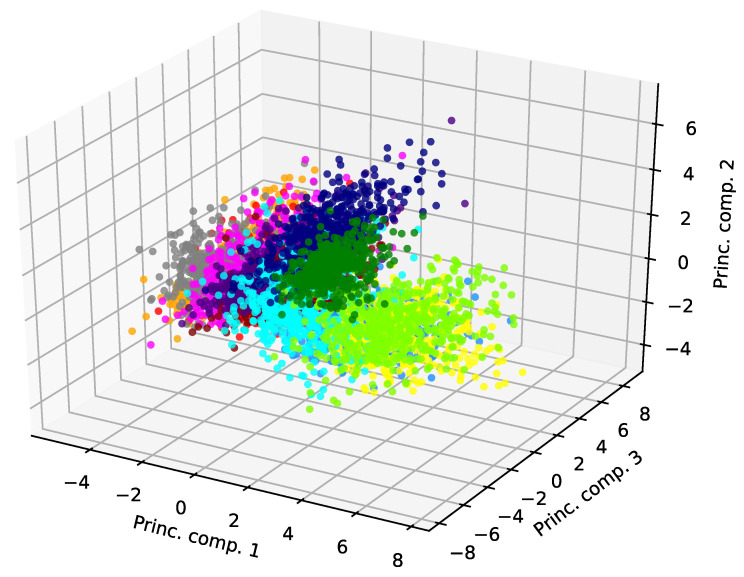
PCA of the entire dataset, where different colors correspond to different students, projected onto the first three components. The percentage of variance explained by the first, second, and third principal component is 35.8%, 14.6%, and 9.5%, respectively.

**Table 1 sensors-22-04857-t001:** Anomaly scores summary.

	Anomaly Score Description	Anomaly Score Range	Detection
PCA	Opposite reconstruction error	$s=-{∥z-\tilde{z}∥}_{2}\in \left(-\infty ,0\right)$	$\hat{y}=\left\{\begin{array}{cc} \mathrm{inlier} & \mathrm{if}\;s>s_{\mathrm{th}}\\ \mathrm{anomaly} & \mathrm{otherwise} \end{array}\right.$
K-PCA	Opposite reconstruction error	$s=-{∥z-\tilde{z}∥}_{2}\in \left(-\infty ,0\right)$	
OC-SVM	Signed distance to the separating hyperplane	$s=\sum_{i}\alpha_{i}K\left(x_{i},z\right)-\rho \in \left(-\infty ,+\infty \right)$	
LOF	Shifted opposite local outlier factor	$s=-\left(lof_{k}\left(z\right)-1.5\right)\in \left(-\infty ,+\infty \right)$	

**Table 2 sensors-22-04857-t002:** Processing time (ms) of the considered anomaly detectors for different PIN lengths *N*.

	PCA	K-PCA	OC-SVM	LOF
N=3	0.104	0.907	0.143	0.623
N=4	0.122	0.931	0.152	0.630
N=6	0.108	0.993	0.165	0.631

## Data Availability

Not applicable.

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
