# Peer review of "Augmented PIN Authentication through Behavioral Biometrics"

_sensors, 2022, doi:10.3390/s22134857_

Round 1

Reviewer 1 Report

This paper proposed a method based on anomaly detection, able to recognize whether the PIN is inserted by the smartphone owner or by an attacker. There are some points should be improved.

1. The core and difficult work should be the acquisition and preprocessing of sensors data. But the current version involves very little in this aspect and needs to be supplemented in detail.

2. The m,s,rad in lines 97,99 and 101 should be in standard form.

Reviewer 2 Report

- The contributions of the paper can be more specific and quantified, e.g. how much is the improvement of the performance by %? What are the originalities? What are the main differences between this papers with other papers, instead of stating the investigation area?

-    Novelty of the work is also a great question? Authors need to justify it.

Challenges, limitations of previous works and research gap are not clearly defined.

-    The paper definitely lacks of clarity, and its organization is not satisfactory.

- The mathematical notation is not clearly explained, and somewhat confusing; several symbols and expressions remain unclear.

-  Authors need to partially modify the abstract with the outcome of the results from the experiment.

-   What is the computation time for the algorithm?Provide running time for the proposed method? Provide the comparison of computation time between the proposed method and other works.     How practicable is your proposed model in real-time?

-   If the proposed method is highly accurate, can you please provide some failure cases for the method? And discuss the underlying cause of these failure cases.

-    It is interesting to see the accuracy of the proposed method in different training sets with different sizes. Please provide the comparison with other works on this issue.

-  There are many algorithm parameters in the proposed method. What's the influence of these parameters?

-  The results are lacking in section experimental results. Apart from the error rate, some other performance indices/ aspects should also be included to test the robustness of the proposed method, e.g. accuracy, means squared error (MSE) etc.

-   To make the paper interesting and be read by the research, including a section of applications of this work. How it is going to give benefit society.

-    "Discussion" section should be edited in a more highlighting, argumentative way. The author should analysis the reason why the tested results is achieved.

- The performance analysis of the system in terms of measurement uncertainty, real time working, and total measurement error should be provided.

Reviewer 3 Report

The manuscript is technically sound and exciting. However, there are several assumptions and limitations in this research work.  The authors acknowledged some but did not state how and whether or not their final research results are affected by them.

Reviewer 4 Report

1. General comments:

The authors proposed a PIN authentication on mobile devices by using behavioral biometrics. The technique verifies whether the correct PIN was inserted by the actual smartphone owner or by an attacker.

The authors exposed various authentication methods that are completely based on behavioral biometrics, and cited 12 references. They proposed a different behavioral biometric authentication method to increase the PIN security. Their method verifies whether the user who typed the PIN was the actual smartphone owner or an unauthorized person. The smartphone movements are recorded during the PIN insertion through built-in motion sensors. Then, an anomaly detection-based system checks whether these movements represent an inlier, or an anomaly in entering the PIN. They implemented the proposed authentication method by using four common anomaly detection algorithms: Principal Component Analysis (PCA), Kernel Principal Component Analysis (K-PCA), One-Class Support Vector Machine (OC-SVM), and Local Outlier Factor (LOF).

2. Analysis:

Quality of the writing: structured coherently

Abstract: Easy to understand

Introduction: well-written, context is clear to some extent.

Problematic: clearly-identified

Method: well-described

Application field: well-identified

Results: well-illustrated.

Conclusion: concise

Related works: more recent references are required

Originality: incremental and very technical (combination of algorithms)

Point 1: More emphasize on originality is required

Point 2: The behavior of users is not necessarily repetitive. Thus, false negatives may happen. In other words, the owner of the smart phone may observe a denial of access. The analysis does not treat false negatives. This analysis is needed.

Round 2

Reviewer 1 Report

The work of this paper is similar to the following paper, but the the description of data collection is not as specific as the following paper, so it is impossible to judge whether the original data is reasonable and reliable.

Reference:

Berend D ,  Bhasin S ,  Jungk B . There Goes Your PIN: Exploiting Smartphone Sensor Fusion Under Single and Cross User Setting[C]// the 13th International Conference. 2018.

Reviewer 2 Report

The authors have correctly addressed the issues I made w.r.t. to the previous version of the manuscript. I thank the authors for their effort in improving the manuscript. The only pending issue is a thorough revision of language. The writing still shows quite a few deficiencies and I strongly recommend a professional proofread before publication

Author Response

We would like to thank the reviewer for the comments and the effort in reviewing our manuscript.

Reviewer 4 Report

The reply is convincing and comments are taken into account

Author Response

(The authors gave the same response as above.)
